# European Database of Explanted UHMWPE Liners from Total Joint Replacements: Correlations among Polymer Modifications, Structure, Oxidation, Mechanical Properties and Lifetime In Vivo

**DOI:** 10.3390/polym15030568

**Published:** 2023-01-21

**Authors:** Miroslav Slouf, Veronika Gajdosova, Jiri Dybal, Roman Sticha, Petr Fulin, David Pokorny, Jesús Mateo, Juan José Panisello, Vicente Canales, Francisco Medel, Alessandro Bistolfi, Pierangiola Bracco

**Affiliations:** 1Institute of Macromolecular Chemistry of the Czech Academy of Sciences, 16206 Prague, Czech Republic; 21st Orthopedics Clinic of the 1st Faculty of Medicine of Charles University and Motol University Hospital, 15006 Prague, Czech Republic; 3Department of Orthopaedic Surgery and Traumatology, Miguel Servet University Hospital, 50009 Zaragoza, Spain; 4Department of Surgery, Medicine School, University of Zaragoza, 50009 Zaragoza, Spain; 5Department of Orthopaedic Surgery and Traumatology, Royo Villanova Hospital, 50015 Zaragoza, Spain; 6Department of Mechanical Engineering-Institute of Engineering Research of Aragon, University of Zaragoza, 50018 Zaragoza, Spain; 7Department of Surgery, Orthopedics and Traumatology, Cardinal Massaia Hospital, 14100 Asti, Italy; 8Chemistry Department and NIS Centre, University of Torino, 10125 Torino, Italy

**Keywords:** ultrahigh molecular weight polyethylene, UHMWPE, total joint replacements, retrieval study, European database, oxidative degradation, micromechanical properties

## Abstract

This contribution lays the foundation for the European database of explanted UHMWPE liners from total joint replacements. Three EU countries (Czech Republic, Italy and Spain) have joined their datasets containing anonymized patient data (such as age and BMI), manufacturer data (such as information on UHMWPE crosslinking, thermal treatment and sterilization), orthopedic evaluation (such as total duration of the implant in vivo and reasons for its revision) and material characterization (such as oxidative degradation and micromechanical properties). The joined database contains more than 500 entries, exhibiting gradual growth, and it is beginning to show interesting trends, which are discussed in our contribution, including (i) strong correlations between UHMWPE oxidative degradation, degree of crystallinity and microhardness; (ii) statistically significant differences between UHMWPE liners with different types of sterilization; (iii) realistic correlations between the extent of oxidative degradation and the observed reasons for total joint replacement failures. Our final objective and task for the future is to continuously expand the database, involving researchers from other European countries, in order to create a robust tool that will contribute to the better understanding of structure–properties–performance relationships in the field of arthroplasty implants.

## 1. Introduction

A recent analysis of Global Burden of Disease (GBD) 2019 data showed that approximately 1.71 billion people globally live with bad musculoskeletal conditions [1], which are also the biggest contributor to disability worldwide. While the prevalence of musculoskeletal conditions increases with age, and it is expected to further increase as the global population ages, younger people are also affected. In the case of joint affections, joint replacements (especially hip, knee, shoulder, ankle and elbow) are widely reported to improve the patients’ quality of life, restoring joint function and reducing pain [2,3,4]. As the number of annual procedures continues to grow all over the world and with the rising costs for public and private health systems, the need to carefully monitor their progress becomes increasingly important.

Starting in the 1970s with the Scandinavian countries, an increasing number of countries have established national arthroplasty registries [5], which are generally aimed at analyzing implant survival and recalling patients in cases of failures. Although they are a powerful source of information for health and regulatory system operators, the registries are generally limited to the collection of patient- and implant-specific data, while they do not provide any analysis or information on the retrieved devices that allows to correlate the characteristics of the materials with their in vivo performances.

Among the components of arthroplasty implants, the ultrahigh molecular weight polyethylene (UHMWPE) liner is the one that has received the most attention in the last 50 years. The constant research for optimized material formulations and processing procedures has made it possible to overcome some critical issues, such as the poor durability of in air radiation-sterilized UHMWPEs that suffer from strong oxidative degradation, which have been progressively replaced by inert-sterilized polyethylene and by two generations of highly crosslinked polyethylene (HXLPE), with an expected improvement in the wear and oxidation resistance [6,7]. The clinical outcome of both historical materials and novel solutions have been the subject of several systematic retrieval studies in the scientific literature [8,9,10,11,12,13,14,15]. However, while numerous and accurate existing studies are far from providing a complete overview of the global situation, for various reasons, including that (i) the methods of analysis and the parameters analyzed are often significantly different from one study to another, making it difficult or impossible to compare different studies; (ii) due to the improved materials and, consequently, longer lifespans, data on modern materials are lacking, especially in some geographic areas, including Europe; (iii) most of the studies include retrievals from North America, Australia or Scandinavian countries only. 

The present project develops precisely from these critical points. Our goal and task for the future is to lay the foundations for a European open database project. The idea stems from collaboration among researchers who have already been active in the field for a long time in the Czech Republic, Italy and Spain and who have shared their respective database of retrievals or suitable portions of them. Our objectives were (i) to explore and/or verify correlations between UHMWPE modifications, structure, oxidative degradation, and lifespan in vivo, even beyond the existing differences in the national practices of each country (different occurrence of certain polyethylene formulations, designs and manufacturers); (ii) to collect data from newer, modern types of UHMWPE liners, the availability of which is still limited, especially in some European countries. In perspective, our goal is to continuously expand the database, involving researchers and structures from other European countries, to create a robust tool that can contribute to monitoring the performance of arthroplasty implants.

## 2. Materials and Methods

### 2.1. Collection of Explanted UHMWPE Liners

The explanted UHMWPE components (retrievals) of total joint replacements (TJRs) come from revision surgery in European hospitals. At the moment, the database contains more than 500 retrievals, coming from the Czech Republic (~350 retrievals), Italy (~100 retrievals) and Spain (~50 retrievals). The great majority of the explants is from the two most frequently replaced joints: hips (~350 retrievals) and knees (~150 retrievals). The project is open, participants from other European countries are welcome and additional information is available online (*https:/mirekslouf.webnode.cz/UHMWPE* (accessed on 20 January 2023) or upon request to any of the three corresponding authors.

### 2.2. Preparation of Testing Specimens from UHMWPE Liners

The preparation of the testing specimens comprised three steps, which are illustrated in Figure 1. In the first step, we prepared a 2 mm thick cross-section through the explanted UHMWPE liner so that it went through both worn and unworn regions (Figure 1a). During the cutting, we could not employ a common mechanical saw, which could heat the material and influence its morphology. The recommended option is to fix the sample firmly and use a very sharp blade from surgical steel. In the second step, we prepared two rectangular samples from worn and unworn regions (Figure 1b). The prisms from the 2 mm sections were easily cut with any sharp knife or scalpel. In the last step, the two rectangular sections were cut with a rotary microtome (such as the RM 2155; Leica, Vienna, Austria) to obtain a 200 μm thin section going through the whole sample (Figure 1c). The 200 μm section was employed in the infrared microspectroscopy (IR; Section 2.3), while the smooth cut surface was used for the determination of the micromechanical properties by means of microindentation hardness testing (MH and MHI; Section 2.4).

### 2.3. Infrared Microspectroscopy

The infrared (IR) data for the characterization of the retrieved UHMWPE liners can be obtained with any IR microscope that can measure 1D line scans (i.e., series of spectra as a function of the distance from the articulating surface). In our case, most of the data were measured with an IR microspectrometer, Thermo Nicolet 6700, with the FTIR microscope Continuum (Thermo Scientific, Brno, Czech Republic). The IR spectra were measured from the 200 µm thick cut sections, as shown in Figure 1c, using the transmission mode by the accumulation of 4 scans with a resolution of 4 cm^−1^. The typical distance between the individual measurements in the line profile was 100 μm. 

Three types of IR indexes were calculated from each spectrum. The oxidation index (OI; proportional to the number of C=O bonds), trans-vinylene index (VI; proportional to the concentration of C=C bonds) and crystallinity index (CI; proportional to the local volume fraction of the crystalline phase). For UHMWPE liners, OI is widely accepted as a measure of oxidative degradation, VI is proportional to the absorbed radiation dose during crosslinking and/or sterilization and CI correlates with the crystallinity determined from differential scanning calorimetry (DSC; [16]). The oxidation index was determined as the ratio of the C=O band area at 1720 cm^−1^ to the standard methylene band area at 1370 cm^−1^ [17] (ASTM F2120):(1)$$\mathrm{OI}=\frac{A_{1720}}{A_{1370}}\;.$$

The trans-vinylene index was calculated as the ratio of the C=C band area at 965 cm^−1^ to the standard methylene band at 1370 cm^−1^ [18,19] (ASTM F2381):(2)$$\mathrm{VI}=\frac{A_{965}}{A_{1370}}\;.$$

The crystallinity index was assessed using the semi-empirical formula CI = CA/(CA + 1), where CA is the ratio of the area of the band at 1897 cm^−1^ (assigned to the PE crystalline phase) to the area of the band at 1303 cm^−1^ (assigned to the PE amorphous phase) [16,18]:(3)$$\mathrm{CI}=\frac{A_{1897}/A_{1303}}{A_{1897}/A_{1303}+1}.$$

The calculated values of OI, VI and CI were combined into OI, VI and CI profiles, as illustrated in Figure 2a. The OI and CI profiles usually exhibit a characteristic camel shape with two peaks corresponding to subsurface oxidation maxima, while the VI profiles are usually flat, as exemplified and discussed elsewhere [18,20]. From each profile, several standardized indexes, such as the maximal values of indexes in peak regions and average values of indexes in the central region, were determined. The explanation of the notation of the indexes is provided in Figure 2b, and a complete list of indexes determined for each sample is summarized in Figure 2c. The calculation of the OI, VI and CI profiles from the series of IR spectra from worn and unworn regions of each sample was standardized and automated by means of our MPINT package (Appendix A). 

### 2.4. Microindenation Hardness Testing

The micromechanical properties of the explanted UHMWPE liners were determined by means of microindentation hardness testing from two locations: the maximum oxidation (around the main peak of the OI profile in Figure 2a) and the central region of the samples (blue region in Figure 2a). The testing specimens for microindentation were the cut surfaces from worn and unworn regions of the samples (light-red prisms in Figure 1c). The micromechanical properties can be obtained with arbitrary noninstrumented or instrumented indentation hardness testers. In this work, most of the data were collected with the noninstrumented microindentation tester VMHT Auto Man (UHL; Asslar, Germany) and the instrumented indentation hardness tester MCT (CSM; Corcelles, Switzerland). Both the noninstrumented microindentation hardness testing (MH) and instrumented microindentation hardness testing (MHI) measurements were carried out using the Vickers method: a diamond square pyramid (with an angle between nonadjacent faces of 136^°^) was forced against the flat surface of a specimen (dwell time = 6 s). For each measured surface, at least 10 independent measurements/indentations were made, and the results were averaged.

The micromechanical property determined from the MH measurements was the Vickers hardness (*H*_v_). The micromechanical properties evaluated from the MHI measurements were the Martens hardness (*H*_M_; also referred to as universal hardness), indentation hardness (*H*_IT_), indentation modulus (*E*_IT_), indentation creep (*C*_IT_) and the elastic part of the indentation work (η_IT_). For semicrystalline polymers including UHMWPE, the values of *H*_V_, *H*_M_ and *H*_IT_ are proportional to each other and to the macroscopic yield stress (*Y*) according to Tabor’s relation (*H* ≈ *H*_V_ ≈ *H*_M_ ≈ *H*_IT_ ∝ 3*Y*), as explained elsewhere [21]. The values of *E*_IT_ are proportional to the macroscale elastic moduli, the values of *C*_IT_ are related to the macroscale creep and the values of *η*_IT_ are defined as the ratio between the elastic and total deformation in the indentation experiment. The calculations of *H*_IT_, *E*_IT_, C_IT_ and n_IT_ were based on the theory of Oliver and Pharr [22]. The exact definitions of the above-listed micromechanical properties can be found in suitable reviews or textbooks dealing with micro- and/or nanoindentation [22,23,24]. A more detailed description of the MHI experiments, together with the illustration that the MHI measurements were reliable and reproducible, can be found also in our recent studies [25,26,27].

In analogy with the IR evaluation, we obtained several values of each micromechanical property, depending on the region and location. The notation for the obtained properties was similar to that of the IR indexes. For example, the measurement of the Vickers microhardness yielded six values: *H*_V_(max,U), *H*_V_(max,W), *H*_V_(max,UW), *H*_V_(ave,U), *H*_V_(ave,W) and *H*_V_(ave,UW)—see the analogous values for OI and CI in Figure 2c.

### 2.5. Summary: Standardized Data Collection and Processing Protocol

In principle, the data collection and processing consisted of four steps. All four steps were standardized in the sense that we defined exact protocols that were followed by all participants in order to achieve reasonable, reliable and reproducible results. Moreover, the key parts of the second step (IR data evaluation) and the fourth step (data mining) were automated by means of Python libraries and scripts, as described below. The four steps in the data collection and processing are provided below.

**Collecting and measuring UHMWPE retrievals** must be performed manually. The retrievals were obtained from the individual hospitals, as described in Section 2.1. The specimens for the IR and microindentation measurements are prepared according to the procedures in Section 2.2. The IR microspectroscopy and micromechanical properties were measured as designated in Section 2.3 and Section 2.4, respectively. **IR data processing** is automated. From each sample, we measured the IR line profiles (a set of IR spectra as a function of the distance from the articulating surface; Section 2.3) and micromechanical properties (which were measured and evaluated from the central region and the region with the maximum oxidation; Section 2.4). The IR spectra were converted automatically to OI profiles, VI profiles and CI profiles by means of our MPINT package (Appendix A). **Storing data** in a well-defined format means must be performed manually. All participants had the same Excel template into which they had to insert information concerning the analyzed explants. The description of each sample included anonymized patient data (such as age, weight and BMI), the manufacturer’s data concerning the UHMWPE material (such as type of crosslinking, thermal treatment and sterilization) and the surgical data (such as total time of the implant in vivo and reasons for revision). Moreover, an experienced user must check the IR microspectroscopy and microindentation data and insert properly averaged and/or maximal values into the database, as described in Section 2.3 and Section 2.4. **Data mining** is automated. From the previous step, the data were stored in Excel files with the defined format. The number of Excel files corresponded to the number of project participants (at the moment, we have three files from the Czech Republic, Italy and Spain). Our MDBASE package can combine (an arbitrary number of) Excel files (on condition that they have the same structure) into one database and create various standard statistical plots and calculations. The MDBASE package is described in Appendix B, and the structure of the database, with which the MDBASE works, is summarized in Appendix C. The MDBASE software was designed to be as intuitive and user-friendly as possible. All statistical plots in the following sections were created with MDBASE, by means of simple Python scripts, which are available upon request to the first author.

## 3. Results and Discussion

### 3.1. Correlation among Oxidation, Crystallinity and Hardness

At the very beginning of our study, it was important to verify that the data collected and processed by three independent teams in three different countries were reasonable, comparable and compatible. For this purpose, we selected the well-established relationship between the oxidative degradation, degree of crystallinity and microhardness of the UHMWPE materials [16,28]. The fact that the oxidative degradation of semicrystalline polymers including UHMWPE results in an increase in their crystallinity and microhardness has been observed and explained in multiple studies. Briefly, the oxidation of polyolefins is connected with chain scissions [29], which occur preferentially on the stressed parts of polymer chains, such as highly constrained entanglements [30], loops on the surfaces of crystalline lamellae [31,32] or tie molecules interconnecting different lamellae [31,33]. These chain scissions are followed by spontaneous *additional crystallization* of freed molecular fragments; the process is also known as *cold crystallization,* because it does not require elevated temperatures [34]. This increase in the crystallinity leads to the higher stiffness of a given semicrystalline polymer, which is manifested by the increase in all stiffness-related properties, such as elastic moduli, yield stress, and hardness [16,35,36].

**Figure 3 polymers-15-00568-f003:**
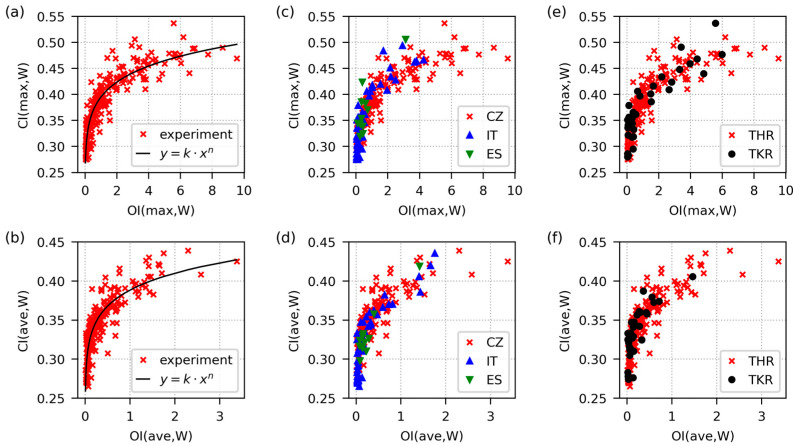
Correlations between oxidation and crystallinity for all evaluated UHMWPE liners in the current version of the database. The data came from worn regions. The upper row (**a**,**c**,**e**) and lower row (**b**,**d**,**f**) show the correlations between the maximum and average values of OI and CI, respectively. The first column (**a**,**b**) displays all data fitted with power law functions. The second column (**c**,**d**) documents that the data from all three countries (CZ = Czech Republic, IT = Italy and ES = Spain) obeyed the same trend. The third column (**e**,**f**) illustrates that the UHMWPE liners from hips and knees (THR = total hip replacement and TKR = total knee replacement) obeyed the same trends as well.

**Figure 4 polymers-15-00568-f004:**
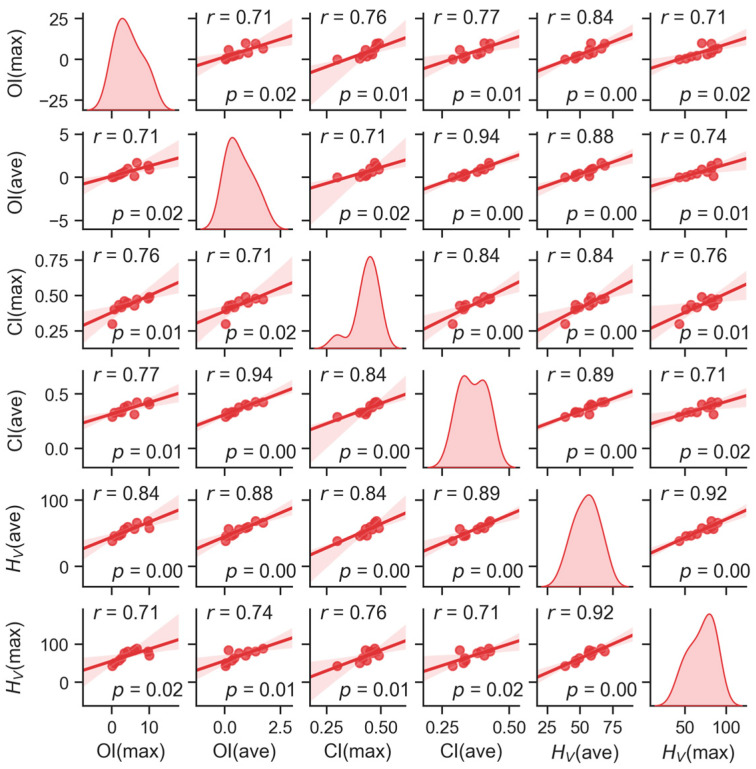
Scatterplot matrix graph showing the correlations between oxidation, crystallinity and microhardness of the explanted UHMWPE liners. All correlated quantities (OI, CI and *H*_V_) are the maxima or averages from both worn and unworn regions, but the index UW is not shown for the sake of graph description clarity (i.e. for example OI(max) and OI(ave) stand for OI(max,UW) and OI(ave,UW), respectively). The main diagonal elements of the graph show the distributions of individual quantities, whereas the off-diagonal elements show the correlations between each pair of quantities (including Pearson’s correlation coefficients, *r*, and *p*-values, which were calculated with the MDBASE program using standard statistical procedures described elsewhere [41]). The translucent bands around the regression lines represent the 95% confidence interval of the regression estimate.

Figure 3 evidences strong correlations among the oxidative degradation and crystallinity of the UHMWPE retrievals. The plots in the upper row of Figure 3 contain the maximum OI and CI values, while the plots in the lower row contain the average OI and CI values. The data came from worn locations, which showed weaker correlations than unworn locations. Therefore, the analogous OI-CI correlations for unworn locations (U) and OI-CI correlations for combined datasets (UW) were even stronger (as proved by further statistical analyses that are discussed below within this section). The fact that the observed trend was in agreement with the theoretical prediction confirmed the correctness of our experimental data. Figure 3a,b illustrate that the crystallinity index (CI), as a function of the oxidation index (OI), obeyed a simple power law relationship (*y* = *k·x^n^*, where *y* = CI, *x* = OI, and *k* and *n* are empirical constants). At low to medium oxidation levels, CI increased with OI fast and almost linearly due to chain scissions and additional crystallization, as discussed in the previous paragraph. At higher oxidation levels, CI increased with OI more slowly, which could be attributed to the strong damage of the polymer chains at all locations, including crystalline lamellae. Figure 3c,d show the same data marked according to the country of origin. The trends for all three countries were very similar, which confirmed the reliability and reproducibility of the data collection, sample preparation, measurements and data processing protocols employed in all three independent laboratories. Figure 3e,f show the data marked according to the replaced joint type. Due to the different geometries of total hip replacements (THRs) and total knee replacements (TKRs), the acting stresses, wear rate and, perhaps, oxidative degradation rate might be different [37,38,39,40]. Nevertheless, similar polymers with similar oxidative damage should exhibit similar crystallinities, regardless of a joint replacement type. This theoretical expectation was fully confirmed by our experimental data.

Figure 4 illustrates the correlation between the structural changes (represented by OI and CI) and the micromechanical properties (represented by the Vickers microhardness, *H*_V_). Unlike the previous figure, the correlated quantities are maximal or average values from both worn and unworn locations. The number of data points is somewhat limited—this is given by the fact that micromechanical characterization has been performed for only a few typical cases. Nevertheless, all correlations between oxidation, crystallinity and microhardness were strong (as proved by the Pearson’s correlation coefficients, *r*, in the upper left corner of the plots in Figure 4) and statistically significant (as evidenced by the *p*-values in the lower right corner of the plots in Figure 4). The values of the Pearson’s *r* ranged from +1 (total positive correlation) through 0 (no correlation) to –1 (total negative correlation). The *p*-values represent the probability that we would observe such a strong (or stronger) correlation just by coincidence (by convention, the correlation is regarded as statistically significant if *p* < 0.05) [41]. The strong oxidation–crystallinity–microhardness correlations are in agreement with the abovementioned literature [16,34,36] and confirmed the reliability of our measurements.

**Figure 5 polymers-15-00568-f005:**
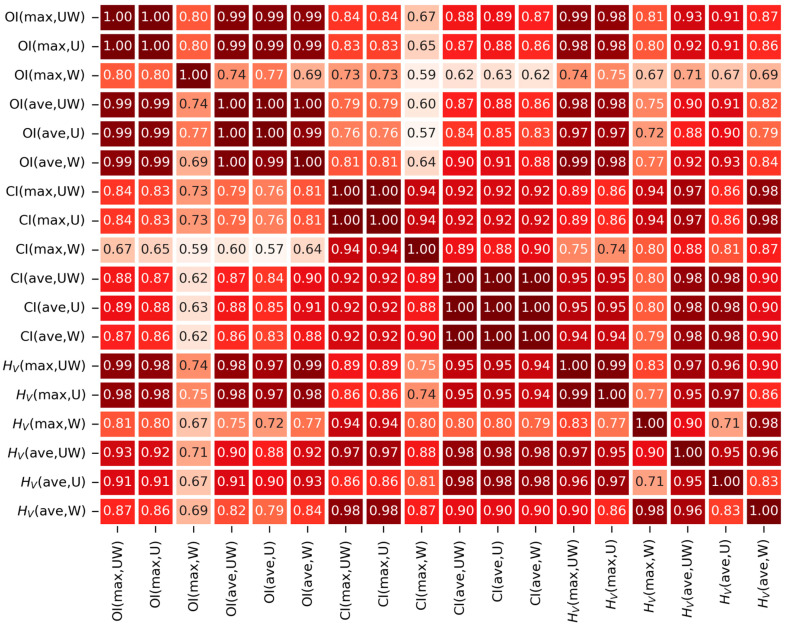
Correlation matrix table showing Pearson’s coefficients, *r*, for all pairs of experimentally determined values of the oxidation indexes (OI), crystallinity indexes (CI) and microhardness (*H*_V_). The table is presented as a heatmap (a darker color means a stronger correlation). The notation and determination of the individual types of OI and CI values are explained in Figure 2.

Figure 5 summarizes all studied oxidation–crystallinity–microhardness correlations in the form of a correlation matrix table. The correlation matrix table displays Pearson’s correlation coefficients, *r*, in the form of a heatmap plot, where the highest *r* values are marked with a dark red color and the lower *r* values with lighter shades of red. The correlation matrix is symmetric with respect to the main diagonal, which contains values equal to 1 (i.e., autocorrelations), while the off-diagonal elements characterize the correlations between all pairs of quantities and prove that all of them are strong (all Pearson’s *r* values are higher than 0.55, and 74% of them are higher than 0.70). Moreover, the matrix shows several groups of properties with almost perfect correlations, mostly located along the main diagonal. These groups are quite logical (as briefly discussed below) and, as a result, they can be regarded as yet another independent confirmation that the measured data are reasonable and reliable. The first small group of strongly correlated properties is the small region OI(max,UW)–OI(max,U) in the upper left corner. Our experience shows that the maximum OI values were found in unworn regions, because the highly oxidized material in worn regions is simply removed due to the mutual motions of the TJR components [33]. Consequently, in most cases OI(max,UW) = OI(max,U), and we obtained (almost) autocorrelation. The second group of strongly correlated properties was the region OI(ave,UW)–OI(ave,W), located around the main diagonal in the upper left area of the matrix. In this case, the strong correlations document the logical fact that oxidative degradation in the central part of a given liner is almost constant in both worn and unworn locations. Moving down along the main diagonal, we found another two strongly correlated regions that confirm the close relationship between oxidative degradation and crystallinity: (i) the region CI(max,UW)–CI(max,U) is analogous to OI(max,UW)–OI(max,U), and (ii) the region CI(ave,UW)–CI(ave,W) is analogous to OI(ave,UW)–OI(ave,W). In the lower right quadrant of the matrix, we can observe strong correlations between crystallinity and hardness, the strongest of which are those between the average crystallinity (CI(ave,UW), CI(ave,U) and CI(ave,W)) and the average values of microhardness (*H*_V_(ave,UW), *H*_V_(ave,U) and *H*_V_(ave,W)). The correlations between the maximal crystallinities and maximal values of hardness were usually somewhat weaker, as the measurement of the microhardness close to the edge of the sample (where we usually observe the maximum oxidation and crystallinity) is less reliable [42]. The weakest correlations were observed for OI(max,W) and CI(max,W), because the highest oxidative degradation in the worn regions is usually very close to the worn surface, and this is somewhat more difficult to detect by IR microscopy due to the presence of edge artifacts. Nevertheless, even the correlations including worn regions were reasonably strong, as documented by the high values of *r* in Figure 5 and illustrated in the scatterplot graphs in Figure 3.

For the sake of completeness, we should briefly discuss the difference between Figure 4 and Figure 5. Both figures are standard types of statistical plots, both can be calculated with the MDBASE package, and both show the correlation between the selected quantities. The difference consists in that the scatterplot matrix graph (Figure 4) contains more information concerning the individual correlations, while the correlation matrix table (Figure 5) can visualize more correlations that are characterized by just a single number (here: Pearson’s *r*). The second important difference, specific to this work, was the fact that Figure 4 was calculated for all cases where *H*_V_ was measured, whereas Figure 5 was calculated without the samples exhibiting the extremely high oxidative degradation (i.e., without the samples with OI(max,UW) > 4, where the scatter of the data increased, as documented in Figure 3). The calculation of Figure 5 with the full dataset provided almost identical results, but the groups of strongly correlating samples were not so clearly visible as the highly oxidized sample, added something similar to a random noise to the calculation. 

### 3.2. Oxidative Degradation vs. UHMWPE Sterilization

The sterilization of the UHMWPE liners with γ-rays (gamma sterilization) was shown to cause higher oxidative degradation in vivo than modern sterilization methods employing ethylene oxide (EtO) or gas plasma [6,43,44,45]. The reason was that the residual radicals from gamma sterilization, which can survive in the polymer for several years, cause long-term oxidative degradation [6,34,46]. In the pre-gamma sterilization era, the hospitals in the Czech Republic sterilized UHMWPE using formaldehyde (HCHO) with remarkable success. According to historical orthopedic records, the HCHO sterilized UHMWPE liners exhibited very long lifetimes in vivo with minimal wear [46]. A few of the HCHO sterilized liners could be retrieved—due to their extremely long lifespan—even in this study.

**Figure 6 polymers-15-00568-f006:**
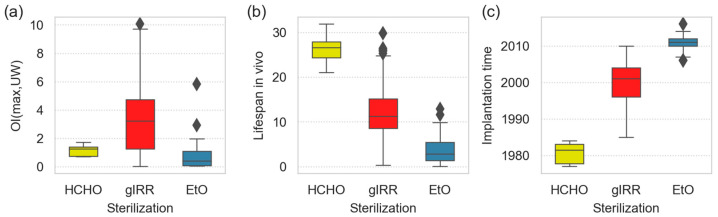
Differences among the UHMWPE liners sterilized by formaldehyde (HCHO, yellow boxes), γ-irradiation (gIRR, red boxes) and ethylene oxide (EtO, blue boxes). The boxplots show (**a**) maximum oxidation index, (**b**) total time in vivo and (**c**) time at which the liner was implanted. All results are shown as boxplots; each box contains five lines, which denote, from bottom to top, the smallest observation (sample minimum), lower quartile (the first quartile, Q1), median (the 2nd quartile, Q2), upper quartile (the 3rd quartile, Q3), and largest observation (sample maximum); single rhombus points denote which observations can be regarded as outliers.

Figure 6 compares the oxidative degradation and performance of UHMWPE liners sterilized with formaldehyde (HCHO; yellow boxes), γ-irradiation (gIRR; red boxes) and ethylene oxide (EtO; blue boxes). Figure 6a documents that the gamma-sterilized UHWPEs exhibited higher oxidative degradation than both the HCHO- and EtO-sterilized liners. The broad range of OI values for the gamma-sterilized samples resulted from the wide variety of samples that were implanted from the 1980s up to the 2010s. On the one hand, some TJRs with the gamma-sterilized samples failed quite early due to non-material-related reasons, such as luxation and, consequently, they exhibited rather low oxidative degradation. On the other hand, some exceptional TJRs with gamma-sterilized UHWPEs survived very long despite heavy oxidation, perhaps due to the lower activity of the patients. In spite of these scattered data, the difference between the gIRR-sterilized and EtO-sterilized samples was statistically significant (*p* = 0.002), and the difference between the gIRR-sterilized and HCHO-sterilized samples was at the edge of statistical significance (*p* = 0.055). Figure 6b seems to suggest that the HCHO-sterilized samples were by far the best from the point of view of in vivo lifespan. However, our retrieval study started approximately in the year 2000, while the last HCHO-sterilized liners were implanted in 1985 (Figure 6c, yellow box). Therefore, the HCHO-sterilized liners, which might have failed earlier, could not be caught within this work. At the same time, the EtO-sterilized liners seemed to exhibit the worst performance in vivo. Here, we face the opposite problem: most of the EtO-sterilized liners were implanted after the year 2010 (Figure 6c, blue box); thus, we had no chance to catch the EtO that survived in vivo for longer durations. As the failed EtO-sterilized liners displayed quite low oxidative degradation (Figure 6a), most of them failed due to the fact of nonmaterial-related reasons, such as luxation or infection. These time-related problems are just one of the general limitations of all retrieval studies, which are discussed in more detail in Section 3.4 below. On the other hand, the average lifespan of the HCHO-sterilized liner was >25 years, which seems to be significantly higher than the minimal theoretical value calculated from straightforward subtraction: 2000 (start of retrieval study) – 1985 (last implants with HCHO-sterilized UHMWPE) = 15 years. This suggests, in agreement with abovementioned studies [6,20], that γ-sterilization was the worst of the three compared sterilization techniques.

### 3.3. Oxidative Degradation vs. Reasons for TJR Failures

The advantage of our database project consists in the fact that it contains not only manufacturer and orthopedic surgery data but also systematically processed data from IR microspectroscopy and micromechanical measurements. This offers an opportunity to correlate orthopedic evaluations (such as the reasons for the TJR failures) with the structural characterization of the failed materials (the most important of which is the oxidative degradation). The close connection between the oxidative degradation and performance of the UHMWPE liners has been observed by many authors and summarized in the UHMWPE handbook by Kurtz and Oral [47], who concluded that reliable predictions are difficult, but the oxidative damage in a given UHMWPE region can be characterized as “low” when OI < 1 or “critical” if OI > 3. The value of the OI should be determined according to the ASTM F2120 standard (like in this work), and it should represent the maximum oxidation index in a given UHMWPE location. The authors of the abovementioned study claim that low oxidation (OI < 1) means that it is very difficult to demonstrate a negative impact on the mechanical properties, whereas critical oxidation (OI > 3) indicates that the ability of the material to withstand long-term mechanical loading in vivo has been compromised.

Figure 7 summarizes the differences in the oxidative degradation of the UHMWPE liners with different reasons of failure. The reasons of failure were inserted into the database by orthopedic surgeons based on their inspection of prerevision roentgenograms, their observations during the revision surgery and their overall evaluation of each individual case. In the next step, the orthopedic description of the reasons for the TJR failure were sorted into four categories:Wear: an aseptic loosening or highly damaged tissues around TJR;Mechanical damage: broken or completely worn UHMWPE liner;Infection: strong local infection around the implanted TJR;Other: all other reasons, such as luxation or poor stability of the implant.

All four of the above listed categories are displayed in Figure 7 and show several interesting features. At first, the trends for the maximum oxidation indexes (Figure 7a) and average oxidation indexes (Figure 7b) were very similar. This is yet another confirmation of the strong correlation between the maximum and average OI (as observed in Figure 4 and Figure 5). In other words, if a UHMWPE liner is oxidized on the surface, it is usually damaged also inside, albeit to a smaller extent. Second, the highest OI were observed for the UHMWPE implants that were damaged mechanically—in most cases, they were broken. This corresponded with the observation of Kurtz and Oral [47] that strong oxidative degradation was connected with compromised mechanical properties. Moreover, the minimal value of the OI(max,UW) for the UHMWPE liners failed due to the fact of mechanical damage (lower line of the red box in Figure 7a) was approximately three (the exact value = 2.98), which is in remarkable agreement with the abovementioned study. Additional interesting features shown in Figure 7 include the wide range of the OIs for the UHMWPE implants that failed due to the fact of wear-related problems (indicating that aseptic loosening may occur even at a low oxidative degradation) and very low OIs of the implants that failed due to the fact of infection (corresponding to the general surgical observation that strong infections around TJR usually occur shortly after the implantation).

Considering the strong correlations between the oxidation and crystallinity (illustrated in Figure 3, Figure 4 and Figure 5), it is possible to calculate boxplots analogous to Figure 7, where the y-axis shows the CI instead of OI. Nevertheless, the observed differences in the crystallinities between the groups were not as clear as in the case of the OI. This confirms that the oxidative degradation was an important reasons for the TJR failures, while the association between crystallinity and TJR failures was weaker.

### 3.4. Limitations of This Study and Current Version of the Database

It has been demonstrated that our UHMWPE database contains enough information to show relevant results. Section 3.1 documents the correlation among the oxidation, crystallinity and hardness of the explanted UHMWPE liners. Section 3.2 evidences the statistically significant differences in the oxidative degradation of the UHMWPEs with different types of sterilization. Section 3.3 illustrates the relationship between oxidative degradation and reasons of TJR failure.

On the other hand, both this study and the current version of the database exhibit some limitations. The limitation of the current database is the lower amount of data from specific types of UHMWPEs. Although >500 explants may seem like a quite sufficient number, the database does not include enough data for highly crosslinked UHMWPEs. This is given by the fact that highly crosslinked UHMWPEs have been introduced to clinical practice quite recently, and their failures are not so frequent. Even if the lack of failed highly crosslinked UHMWPEs in the database can be regarded as indirect proof of their higher average lifetimes in vivo, the statistically significant difference between non-crosslinked and crosslinked UHMWPEs cannot be demonstrated at the moment.

**Figure 8 polymers-15-00568-f008:**
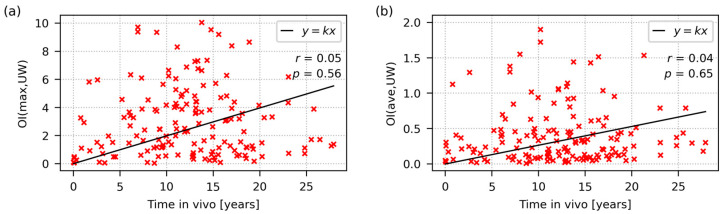
Increase in the oxidative degradation of UHMWPE implants as a function of their in vivo lifetime: (**a**) maximum oxidation index; (**b**) average oxidation index from both worn and unworn locations. The low values of the Pearson’s correlation coefficients, *r,* in both plots evidence the negligible correlations between the OI and in vivo lifetime. The high *p*-values document that the correlations were statistically insignificant.

The general limitation of the whole study consists in that any database of UHMWPE retrievals—regardless of its size—cannot show and/or confirm all expected correlations simply due to the fact that some trends are not so clear, straightforward and/or unambiguous. For instance, we might expect that oxidative degradation will increase with the average lifetime of the UHMWPE liner in vivo. Figure 8 documents that this correlation is rather weak, even if the dataset’s size is large. One of the reasons for the weak correlation between the average lifetime and oxidation is shown in Figure 6. Some types of UHMWPE liners (such as unmodified UHMWPEs sterilized with formaldehyde, which are represented by the yellow boxes in Figure 6) exhibited a very high lifespan with minimal oxidative degradation. Similarly, unmodified UHMWPE liners sterilized with EtO, which are quite common in Italy, are reported to perform very well [17,48]. In this context, it is worth noting that the analogy between HCHO sterilization and EtOH sterilization was confirmed by a recent study [46]. Nevertheless, some other types of UHMWPE liners (especially those sterilized by gamma radiation) were heavily oxidized and failed quite early, as documented by the experimental data in the upper left areas of Figure 8a,b. It is possible that multifactor analysis and multiple regression methods might reveal and/or confirm further relationships if the database size increases even more in the future—this will be the subject of our further research.

## 4. Conclusions

We collected and characterized >500 explanted UHMWPE components from total joint replacements. The retrieved UHMWPE components came from major hospitals in three European countries: Czech Republic, Italy and Spain. The description of each sample comprises anonymized patient data (such as age and BMI), manufacturers data (information concerning UHMWPE modifications), surgical data (namely, reasons for the implant revision) and characterization of the material (obtained by standardized protocols for the processing of IR microspectroscopy and microindentation data).

In the first step of this study, we verified the reliability and mutual compatibility of the collected data by means of well-established correlations among oxidation, crystallinity and hardness (Section 3.1). In the second step, we investigated the relationship between the sterilization and oxidative degradation of the UHMWPE liners (Section 3.2). Moreover, we managed to demonstrate some relationships between UHMWPE properties and reasons for TJR failures according to orthopedic evaluation (Section 3.3). Finally, we identified and discussed some limitations of our study, which were connected either with the limited database size or with intrinsic restrictions of the retrieval studies (Section 3.4).

The final objective of our future work is to create a large, complete and robust database of failed UHMWPE liners from EU countries. Such a database could be used as a tool for unbiased, noncommercial evaluations of various UHMWPE materials for TJR for more reliable predictions of TJR lifespan and for better understanding of structure–properties–performance relationships in the field of UHMWPE implants. The project is opened to participants from other countries, who can contact anyone of the three corresponding authors in order to become members of the team.

## Figures and Tables

**Figure 1 polymers-15-00568-f001:**
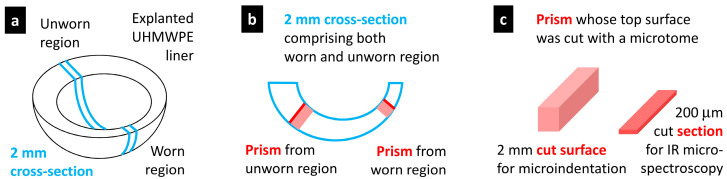
Scheme of the sample preparation: (**a**) The explanted UHMWPE liner was cut so that we obtained a 2 mm thick cross-section, which went through both worn (damaged) and unworn (nondamaged) regions of the implant. (**b**) The 2 mm cross-section was used for the preparation of two prisms that spanned from the articulating to the opposite surface of the liner; the first and second prisms were cut from the worn and unworn location, respectively. (**c**) The top surface of both prisms was cut with a rotary microtome in order to obtain 200 μm sections for the subsequent IR measurements (red lines and plates in (**b**,**c**), respectively) and smooth cut surfaces for microindentation measurements (pink rectangles and prisms in (**b**,**c**), respectively).

**Figure 2 polymers-15-00568-f002:**
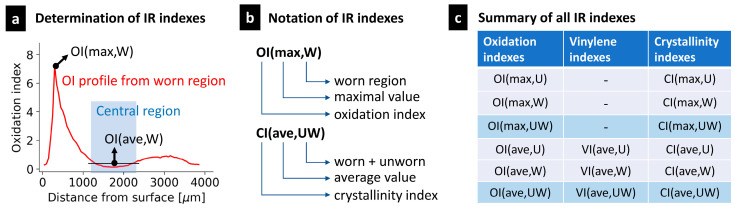
Standardized oxidation indexes (OI), trans-vinylene indexes (VI) and crystallinity indexes (CI), which were determined for each explanted UHMWPE liner in this work: (**a**) a typical OI profile and determination of the maximum and average values of OI; (**b**) explanation of the abbreviations that were used for the determined indexes; (**c**) complete list of indexes, which were determined for each UHMWPE liner. The total number of indexes was 15. The abbreviations U and W stand for unworn and worn regions, respectively. The abbreviation UW represents the combined data from the U and W regions. Therefore, all indexes (OI, VI and CI) were calculated from the three different datasets (U, W or UW). The maximum values of the trans-vinylene indexes (i.e., VI(max,U), VI(max,W) and VI(max,UW)) were not determined, as the VI profiles were approximately constant.

**Figure 7 polymers-15-00568-f007:**
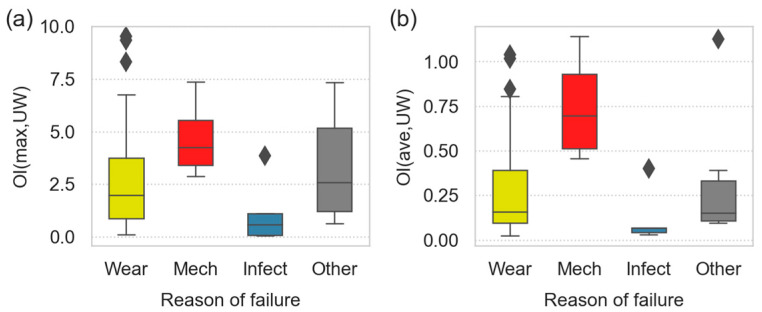
Differences in the oxidative degradation for the UHMWPE liners which failed due to the fact of wear (yellow boxes), macromechanical damage of the implant (red boxes), infection (blue boxes) or other reason (gray boxes). The differences are shown for both the (**a**) maximum and (**b**) average values of OI. The meanings of the box lines and single points are the same as in Figure 6.

## Data Availability

Anonymized data are available upon request to the corresponding authors.

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
