# Peer review of "European Database of Explanted UHMWPE Liners from Total Joint Replacements: Correlations among Polymer Modifications, Structure, Oxidation, Mechanical Properties and Lifetime In Vivo"

_polymers, 2023, doi:10.3390/polym15030568_

Round 1

Reviewer 1 Report

Please correct some misspellings like in row 161 of the manuscript "nanoindentation"

Please make more comments about the reproducibility of microindentation tests

Author Response

Reviewer’s comments and our point-by-point answers

Point-by-point answers to all reviewer’s comments follow. All changes in the manuscript were made with MS Word/Revision tool, so that they could be tracked and visualized easily.

(1) Please correct some misspellings like in row 161 of the manuscript "nanoindentation".

Answer: We re-checked the manuscript and corrected misspellings as the reviewer suggested. Nevertheless, the term [Microindentation] in line 161 is correct. The micromechanical properties in this study were performed by microindentation and not nanoindentation.

(2) Please make more comments about the reproducibility of microindentation tests.

Answer: The comment was inserted in the second paragraph of section 2.4. Briefly, the microindentation measurements are highly reliable and reproducible, as evidenced in our previous publications, which are cited in the inserted text.

Reviewer 2 Report

1. This is an invaluable contribution in providing a sound database with information for continued scientific research in orthopedic liners and other implant information for Total Joint Replacement (TJR).

2. The effort to pool data from various countries and develop this database is lauded.

3. Although the total number of data points n = 500 is well documented, please include the individual contributions from the Czech Republic, Italy and Spain. Please also include the data about how many TKRs and THRs.

4. A schematic diagram in the  line of Figure 1 for TKR would be valuable.

5. An added explanation as to why the relationship between CI and OI should be best described by a power law will be useful. Are k and n merely mathematical constants or is there a materials-based connotation?

6. The "heat map" in Figure 5 is an excellent representation of the experimental data.

7.  What is the exact value = ? (it is missing in line 445.

8. One would tend to imagine that while the time of duration of implant is of particular importance, this itself should be a function of the load experienced by the liner, actual time of use, peri-implant environment etc.  Thus a multiple regression for the data in Figure 8 may be more appropriate.

9. Does the origin of fabrication/manufacturer affect the data?

Author Response

Reviewer’s comments and our point-by-point answers

Point-by-point answers to all reviewer’s comments follow. All changes in the manuscript were made with MS Word/Revision tool, so that they could be tracked and visualized easily.

  1. This is an invaluable contribution in providing a sound database with information for continued scientific research in orthopedic liners and other implant information for Total Joint Replacement (TJR). The effort to pool data from various countries and develop this database is lauded.

Answer: We thank the reviewer for his/her positive evaluation.

  1. Although the total number of data points n = 500 is well documented, please include the individual contributions from the Czech Republic, Italy and Spain. Please also include the data about how many TKRs and THRs.

Answer: The required information was inserted in the 2nd sentence of the 1st paragraph of section 2.1.

  1. A schematic diagram in the line of Figure 1 for TKR would be valuable.

Answer: It is true that Figure 1 shows THR as an example. Nevertheless, for TKR the sample preparation is exactly the same, only the shape of the initial UHMWPE liner is different (acetabular cup vs. tibial plateau). Therefore, if we added Figure 2 in line with Figure 1, it would be almost redundant: Just Figures 1a and 2a would show different shapes of the explanted liners, Figures 1b and 2b would be almost the same and Figures 1c and 2c would be identical.

  1. An added explanation as to why the relationship between CI and OI should be best described by a power law will be useful. Are k and n merely mathematical constants or is there a materials-based connotation?

Answer: The following explanation was added to the revised manuscript: “Figures 3a–b illustrate that the crystallinity index (CI) as a function of oxidation index (OI) obeyed a simple power law relation (y = k⋅x^n, where y = CI, x = OI, and k, n are empirical constants). At low to medium oxidation levels, CI increased with OI fast and almost linearly due to chain scissions and additional crystallization, as discussed in the previous paragraph. At higher oxidation levels, CI increased with OI more slowly, which could be attributed to the strong damage of the polymer chains at all locations, including crystal-line lamellae.”

  1. The "heat map" in Figure 5 is an excellent representation of the experimental data.

Answer: We agree with the reviewer. This type of plot is not too common in materials science, but well-established in the field of statistics, when comparing correlations between multiple quantities.

  1. What is the exact value = ? (it is missing in line 445.

Answer: We thank the reviewer for careful reading. The missing value was inserted in the revised manuscript.

  1. One would tend to imagine that while the time of duration of implant is of particular importance, this itself should be a function of the load experienced by the liner, actual time of use, peri-implant environment etc. Thus a multiple regression for the data in Figure 8 may be more appropriate.

Answer: The reviewer is right. Figure 8 just illustrates that single regression does not show reasonable results at the moment. Advanced statistical methods (such as multi-factor analysis and multiple regression) will be a subject of our further studies when the database size grows. This information was added to the revised manuscript (at the end of the last paragraph of section 3.4).

  1. Does the origin of fabrication/manufacturer affect the data?

Answer: At the moment we have no indication that the same types of UHMWPE (such as neat UHMWPE sterilized with EtO) exhibit different properties depending on the manufacturer. This is logical, as all manufacturers should follow standardized modification protocols.

Reviewer 3 Report

Attached

Author Response

Reviewer’s comments and our point-by-point answers

Point-by-point answers to all reviewer’s comments follow. All changes in the manuscript were made with MS Word/Revision tool, so that they could be tracked and visualized easily.

The authors have presented an interesting work that lays the foundation of the joined-European (Czech Republic, Italy, and Spain) database of explanted UHMWPE liners from total joint replacements containing over 500 anonymized patient data, manufacturers data (i.e., information on UHMWPE crosslinking, thermal treatment, and sterilization), the orthopedic evaluation (i.e., total time of the implant in vivo and reasons for its revision) and material characterization (i.e., oxidative degradation and micromechanical properties). The joined database exhibits strong correlations between the UHMWPE oxidative degradation, degree of crystallinity and microhardness, statistically significant differences among UHMWPE liners with different types of sterilization, and realistic correlations between the extent of oxidative degradation and the observed reasons of total joint replacement failures. However, the authors need to significantly revise the manuscript before resubmitting. The authors may consider fixing the following specific points to improve the manuscript:

Answer: We thank the reviewer for his evaluation and comments, which helped us to improve the quality of our manuscript. Point-by-point answers to all reviewer’s comments follow. All changes in the manuscript were made with MS Word/Revision tool, so that they could be tracked and visualized easily. The exception are the newly added references, which were inserted to Introduction as the reviewer requested. We inserted the new references using Zotero software with Revisions switch off in MS Word.

  1. The authors may like to revise the literature review. The latest literature included in the Introduction section is from 2019.

Answer: Five new references from years 2020-2022, illustrating the importance of the manuscript topic, were added as the reviewer suggested.

  1. The authors should have discussed more about the Python packages in the Method section instead of putting everything in the appendices.

Answer: In principle, the reviewer is right. However, the description of the developed packages (MPINT and MDBASE) is mostly a technical issue (that is why it was put in Appendices) and detailed description of the installation, documentation, and usage of the packages is even more technical issue (and that is why it was put at www-sites of the packages, so that all potential users of our software could find the documentation together with source files and installation instructions – this is a standard way and we employed standard sites/tools such as pypi.org and github.com). In order to address the reviewer comment, we improved the description of both packages at www (as can be verified at the relevant sites: https://pypi/org/project/mpint and https://pypi.org/project/mdbase).

  1. In Figure 2c: The authors didn’t specify what the short forms U, W and UW stand for. W=worn region, UW=unworn region, but what is U?

Answer: We thank the reviewer for careful check of our figures. The description of U, W and UW symbols was somewhat ambiguous. Therefore, we added the following text to the legend of Figure 2: “The abbreviations U and W stand for unworn and worn regions, respectively. The abbreviation UW represents combined data from U and W regions. Therefore, all indexes (OI, VI, and CI) were calculated from the three different datasets (U, W or UW).”

  1. Lines 259-260: The upper and lower rows should be replaced by actual figure numbers such as Fig 3 (a), (b), (c) etc.

Answer: Corrected as the reviewer suggested.

  1. Line 261: ‘that’ should be replaced by ‘than’

Answer: Corrected as the reviewer suggested.

  1. Figure 3 presents the data from the worn regions only. How the authors compare the oxidative degradation and crystallinity correlation of the worn regions with that of the unworn regions (Lines 261-263).

Answer: This is explained directly in the text of the manuscript: Figure 3 illustrates the correlations for unworn regions, while the following Figures 4 and 5 quantify all observed correlations for UW regions and all three regions, respectively. The text of the revised manuscript was slightly modified so that this was clearer.

  1. Lines 266-267: This sentence should be rewritten as ‘CI stabilizes at higher OI’ or ‘CI does not change much at higher OI’.

Answer: We believe that the original phrase “the lower increase in CI at higher oxidation levels” is correct and more precise.

  1. Line 272: ‘Figures 3d-e’ should be replaced by ‘Figures 3e-f’.

Answer: Corrected (we thank the reviewer for his/her careful check).

  1. Line 292: “finished’ must be replaced by an appropriate word.

Answer: The word “finished” was replaced by “performed”.

  1. The y-axis title of Figure 6b could be replaced by ‘Lifespan in vivo’.

Answer: The figure was modified as the reviewer suggested.

  1. The title of Section 3.3 ‘3.3. Oxidative degradation vs. reason of TJR failure’ could be replaced by ‘Oxidative degradation as the reason of TJR failure’.

Answer: The reviewer is right that the title was not correct. However, we modified in in a slightly different way and it reads: “Oxidative degradation vs. reasons of TJR failures”. This is more suitable because section 3.3 compares oxidative degradation of various groups of TJR’s, which exhibited different reasons of failure according to orthopedic evaluation.

  1. The x-axis title of Figure 7: ‘Infect of failure’ could be replaced by ‘Infection’.

Answer: This is a misunderstanding. Figure 7 shows four groups (Wear, Mech, Infect and Other) and title of X-axis (Reasons of failure). Therefore, there is not “infect of failure” label. We modified the graph so that the pad between the group labels and X-axis label was a bit higher.

  1. Line 439: ‘it is usually damaged also inside’ could be replaced by ‘it also usually damages from inside’.

Answer: We decided to keep the original phrase, which is more precise.

  1. Figure 7: The authors should include two more plats for OI (max, W) and OI (ave, W).

Answer: The reviewer is right, but the suggested plots would show basically the same thing. That is why we presented, for the sake of clarity and brevity, just the final plots where both regions (U and W) are combined into one (UW).

  1. Crystallinity index (CI,max and CI,ave) could also be analyzed for the reason of failures (wear, mechanical failure, infection and others).

Answer: The reviewer is right, and we have already performed such analysis, but the association between crystallinity and reasons of failure was weaker. The following brief paragraph, answering the reviewer’s comment, was inserted at the end of section 3.3. of the revised manuscript: “Considering the strong correlations between oxidation and crystallinity (illustrated in Figures 3–5), it is possible to calculate boxplots analogous to Figure 7, where the Y-axis shows CI instead of OI. Nevertheless, the observed differences in crystallinities between the groups are as not clear as in the case of OI. This confirms that the oxidative degradation is one of the important reasons of TJR failures, while the association between the crystallinity and TJR failures is weaker.”

  1. Figure 8: The authors may like to add r values in both plots. The r values will demonstrate the weaker correlation.

Answer: The Pearson’s coefficients (r-coefficients) were added to Figure 8 as the reviewer suggested. Moreover, we added p-values. The close-to-zero values of r-coefficients document negligible correlation. The high p-values document that the observed weak correlations are also statistically insignificant. Legend of Figure 8 was modified accordingly.

  1. Line 488: ‘hundred’ is redundant.

Answer: Removed as the reviewer suggested.

  1. Line 503: ‘The final objective of our work is to create a large, complete and robust database’ is misleading. If this was your objective for this work, you should have achieved it and would have claimed it in this manuscript. Rather, creating a large, complete and robust database is your FUTURE goal. This should be discussed as future direction or future work. This comment is also applicable for Abstract and for the last paragraph of Introduction.

Answer: The Abstract, Introduction and Conclusion were slightly modified as the reviewer suggested. In all three abovementioned parts of the revised manuscript, we mentioned that the creating of large database is our objective and task for the future. The changes are in the last sentence of the Abstract, at the beginning of the last paragraph of Introduction, and in the first sentence of the last paragraph of Conclusion.

Reviewer 4 Report

The article reports a very extensive study about the degradation of UHMWPE in vivo.

I have just few suggestions

Line 100 please write replace IR by infrared

Iine 137 Please write what  DSC means

Line 175 please provide the indentation dwell time

Please Explain better your abbreviations, for example what U means in table of figure 2

Author Response

Reviewer’s comments and our point-by-point answers

Point-by-point answers to all reviewer’s comments follow. All changes in the manuscript were made with MS Word/Revision tool, so that they could be tracked and visualized easily.

The article reports a very extensive study about the degradation of UHMWPE in vivo. I have just a few suggestions:

  1. Line 110 please write replace IR by infrared.

Answer: Replaced as the reviewer suggested.

  1. Line 137 Please write what DSC means.

Answer: DSC stands for differential scanning calorimetry. The full name of the method was added to the revised manuscript.

  1. Line 175 please provide the indentation dwell time.

Answer: Dwell times were 60 s; the required information was added to the revised manuscript.

  1. Please Explain better your abbreviations, for example what U means in table of figure 2.

Answer: The legend of Figure 2 was modified so that this was clearer (this was required by other reviewers as well).

Round 2

Reviewer 3 Report

Authors' responses to my comments have been accepted.